# Artificial Tears: Biological Role of Their Ingredients in the Management of Dry Eye Disease

**DOI:** 10.3390/ijms23052434

**Published:** 2022-02-23

**Authors:** Marc Labetoulle, Jose Manuel Benitez-del-Castillo, Stefano Barabino, Rocio Herrero Vanrell, Philippe Daull, Jean-Sebastien Garrigue, Maurizio Rolando

**Affiliations:** 1Service d’Ophtalmologie, Hôpital Bicêtre, Assistance Publique-Hôpitaux de Paris, 94270 Le Kremlin Bicêtre, France; marc.labetoulle@aphp.fr; 2Department of Ophthalmology, Hospital Clinico de Madrid, Universidad Complutense, 28040 Madrid, Spain; benitezcastillo@gmail.com; 3Centro Superficie Oculare e Occhio Secco, ASST Fatebenefratelli-Sacco, Ospedale L. Sacco, Università di Milano, 20157 Milan, Italy; stebarabi@gmail.com; 4Research Group (UCM 920415), Innovation, Therapy and Pharmaceutical Development in Ophthalmology (InnOftal), Faculty of Pharmacy, Complutense University, 28040 Madrid, Spain; rociohv@farm.ucm.es; 5Ophthalmic Innovation Center, Santen SAS, 91058 Evry, France; daull.phaa@laposte.net; 6Ocular Surface Centre, ISPRE (Instituto di Medicina Oftalmica) Ophthalmic, 16129 Genoa, Italy; maurizio.rolando@gmail.com

**Keywords:** tear film, artificial tears, tear substitutes, ingredients, excipients, dry eye disease, ocular surface, cornea

## Abstract

Dry eye disease (DED) is the most common ocular surface disease, characterized by insufficient production and/or instability of the tear film. Tear substitutes are usually the first line of treatment for patients with DED. Despite the large variety of tear substitutes available on the market, few studies have been performed to compare their performance. There is a need to better understand the specific mechanical and pharmacological roles of each ingredient composing the different formulations. In this review, we describe the main categories of ingredients composing tear substitutes (e.g., viscosity-enhancing agents, electrolytes, osmo-protectants, antioxidants, lipids, surfactants and preservatives) as well as their effects on the ocular surface, and we provide insight into how certain components of tear substitutes may promote corneal wound healing, and/or counteract inflammation. Based on these considerations, we propose an approach to select the most appropriate tear substitute formulations according to the predominant etiological causes of DED.

## 1. Introduction

Dry eye disease (DED) is a condition that may affect between 5% and 50% of the population, depending on age, sex and ethnicity [1]. According to the TFOS DEWS II report, DED is defined as “ocular surface disease characterized by a loss of homeostasis of the tear film, and accompanied by ocular signs, in which tear film instability and hyperosmolarity, ocular surface inflammation and damage, and neurosensory abnormalities play etiological roles” [1]. This disease causes ocular discomfort, fatigue and visual disturbance, which significantly affect the quality of life of patients [1]. Among the different treatment strategies available to patients, the topical administration of tear substitutes, also called artificial tears (even though none of the currently available options successfully recreate the normal human tear film), has been shown as a safe approach to supplement (at least partially) the natural tear film of patients with DED.

A multitude of tear substitutes are currently available on the market worldwide, with a wide variety of ingredients. Most of these tear substitutes are classified as medical devices (MD) or medicines that can be purchased over-the-counter, unlike prescription medicines. In 2016, a Cochrane systematic review highlighted the inconsistencies in clinical trials assessing the safety and performance of tear substitutes, and therefore the inability to compare one to another [2]. This study also showed that the tear substitutes preferred by a patient do not consistently match with the ATs providing the greatest improvement on objective clinical signs. Therefore, the exact properties that drive both the efficiency and the tolerance of ATs are still circumscribed and, in most cases, reported by industry-driven studies. In May 2021 in Europe, a new regulation on MDs (Regulation (EU) 2017/745) was established in order to improve and standardize the evaluation of safety and performance of MDs [3]. This regulation should bring about a better understanding of the properties of the different types of tear substitutes, and more specifically the role of their components. That could help physicians to determine the most suitable tear substitute for each DED patient, according to the clinical findings and the underlying components of the DED in his specific case.

Recently, an updated classification of tear substitutes has been described by *Barabino* et al. and consists of three categories based on their degree of interaction with the eye: wetting agents, multiple-action tear substitutes and ocular surface modulators [4]. While wetting agents only lubricate the ocular surface with a limited residence time, multiple-action tear substitutes can improve tear film quality and quantity without interacting with the ocular surface. As the name suggests, ocular surface modulators interact with and influence the ocular surface in order to counteract DED signs.

In this review, we will first describe the physiopathology of DED and especially its underlying causes. Then, we will review the preclinical and clinical outcomes of the variety of ingredients composing tear substitutes, and we will describe how these ingredients may target one or several of the underlying mechanisms of DED.

## 2. Physiopathology of Dry Eye Disease

The tear film acts as a barrier to protect the ocular surface from the external environment. The major part (in terms of volume) of tear film is the aqueous layer (TFAL) produced by the lacrimal glands. It is composed of water, electrolytes, metabolites and proteins, which lubricate the ocular surface and maintain its healthy osmolarity (Figure 1A). Produced by the Meibomian glands located in the eyelids, a tear-film lipid layer (TFLL) covers the aqueous layer, protects it from desiccation and helps in its stabilization [5]. The TFLL is made of both polar and non-polar lipids separated into two layers [6]. While the non-polar lipid layer is located at the interface with the environment (air), the polar lipid layer stabilizes the TFLL on the TFAL (Figure 1B). Below the aqueous layer is a mucin layer, composed itself in three categories of mucins, i.e., transmembrane, soluble and gel-forming. Transmembrane mucins are present on the surface of epithelial surface cells and form the glycocalyx, which helps to anchor the aqueous part of the tear film on the ocular surface epithelial cells [7]. Soluble and gel-forming mucins are located in the aqueous layer of the tear film, and are responsible for the pseudoplastic behavior of the tear film [7].

Several underlying causes of DED have been identified along with complex biological dysfunction mechanisms that were the basis for the “vicious circle” concept, first introduced by Rolando et al. [8], then well theorized by Baudouin (with detailed schematic representation) [9], hereby quite simplified in Figure 1C. DED has been initially classified into two main subtypes based on the predominant etiology: evaporative dry eye (EDE) and aqueous-deficient dry eye (ADDE) [10]. EDE is induced either by a Meibomian gland dysfunction (MGD) resulting in a lipid layer deficiency or by a mucin dysfunction altering tear film stability. ADDE is due to aqueous layer underproduction. A study showed EDE is three times as common as ADDE in patients with DED [11,12]. However, it is now widely acknowledged that 30% of the patients presented with both EDE and ADDE signs, indicating the existence of a continuum between these subtypes [11]. Recently, the classification has been updated by Asia Dry Eye Society into three categories: increased evaporation (EDE), aqueous deficiency (ADDE) and decreased wettability [13]. The decreased wettability of the cornea and conjunctiva refers to a membrane-associated mucin dysfunction or deficiency which can lead to instability of the tear film and shortened tear film break-up time (e.g., “short break-up time” patients as observed in visual display terminal workers) [13].

Therefore, patients can present different types of DED, with related deficiencies of components of the tear film and abnormalities in the ocular surface epithelium, which should be considered in the management of their condition, and more specifically at the time of choosing the tear substitutes the most appropriate for each specific case.

## 3. Types and Roles of Ingredients Used in Tear substitutes

The main types of ingredients used in the composition of tear substitutes are viscosity-enhancing agents, electrolytes, osmo-protectants, oily agents, antioxidants and preservatives. More interestingly, some roles on wound healing and inflammation have been identified for certain ingredients.

### 3.1. Viscosity-Enhancing Agents

Viscosity-enhancing agents represent the most frequently used ingredients, composing the bulk of ATs. Referred to as demulcents or lubricants as well, the U.S. Food and Drug Administration (FDA) recognizes six categories of these agents that can be used in over-the-counter formulations including cellulose derivatives, dextran, gelatin, liquid polyols, polyvinyl alcohol and povidone [14]. These agents have been proved to be beneficial for patients with DED by increasing the tear film thickness and by increasing retention of ATs at the ocular surface [15]. Moreover, these compounds act as water retaining agents (i.e., hygroscopic properties) which allow them to moisturize the ocular surface by preventing the loss of water. Among the viscosity-enhancing agents used in the ATs formulations, sodium carboxymethylcellulose (CMC), a cellulose derivative from plants, is the most frequently utilized (Table 1). CMC has been shown to be beneficial for patients with mild to moderate DED by improving the corneal surface wettability and tear film integrity [16]. Other viscosity-enhancing agents are used in marketed ATs including hydroxypropyl methylcellulose (HPMC), carbomer, hyaluronic acid (HA), polyvinyl alcohol, povidone, dextran and hydroxylpropyl-guar (HP-guar) (Table 1) [15,17]. These agents are known to be mucoadhesive and mucomimetic due to their branched structure similar to mucin 1, a mucin formed by the goblet cells and playing a protective role of the ocular surface [18,19,20]. A clinical study showed that HA or tamarind seed polysaccharide, another viscosity-enhancing agent, can improve the residence time of ATs on the human ocular surface and reduce DED symptoms [21]. Therefore, viscosity-enhancing agents can be used to replenish and help to maintain the mucin layer in case of DED caused by mucin deficiency.

The source and properties of these agents (e.g., molecular weight) vary and these are able to influence their interaction with the ocular surface (Table 2). If no difference has been observed between these agents in terms of efficacy on the DED signs [22], some studies showed that the combination of CMC and HA improved ocular signs, compared with CMC or HA alone, both on mice and clinically (Figure 2A) [23,24]. Similar outcomes were observed with a combination of HA with HP-guar [25].

Some of these viscosity-enhancing agents can undergo a solution-gelation (sol-gel) transition upon contact with the ocular surface, such as carbomer or HP-guar [20]. Administered as a liquid on the eye, these “in situ gelling systems” form a gel when mixed with tears. Various mechanisms can trigger the sol-gel transition; while HP-guar increases viscosity in contact with borates and divalent ions from tears, carbomer increases viscosity at the pH of the ocular surface [20,26].

These results suggest that viscosity-enhancing agents could help to increase the residence time of ATs on the ocular surface and compensate, at least partially, the mucin layer in case of deficiency. Moreover, there are clues suggesting that combinations of these agents can improve the performance of ATs.

**Table 1 ijms-23-02434-t001:** Examples of marketed artificial tears composed only of viscosity-enhancing agents and electrolytes. The composition excludes water and hydrochloric acid/sodium hydroxide for pH adjustment.

Brand Name	Viscosity-Enhancing Agents	Electrolytes	Others	References of Clinical Studies
Refresh^®^ Classic^®^ (Allergan, Irvine, CA, USA)	Povidone, PVA	Sodium chloride		
Bion^®^ Tears (Alcon, Fort Worth, TX, USA)	Dextran 70, HPMC	Calcium chloride, magnesium chloride, potassium chloride, sodium bicarbonate, sodium chloride, zinc chloride		[27]
Refresh^®^ Plus^®^ (Allergan, Irvine, CA, USA)	CMC	Calcium chloride, magnesium chloride, potassium chloride, sodium chloride, sodium lactate		[27]
TheraTears^®^ (Akorn Pharmaceutical, Lake Forest, IL, USA)	CMC	Calcium chloride, magnesium chloride, potassium chloride, sodium bicarbonate, sodium chloride, sodium phosphate, boric acid, sodium borate.		NCT02014922 (data not published)
Systane^®^ (Alcon, Fort Worth, TX, USA)	PEG 400, Propylene Glycol, HP-guar	Calcium chloride, magnesium chloride, potassium chloride, sodium chloride, zinc chloride, boric acid	Preservative: PQ (POLYQUAD^®^)	[28]
Blink^®^ Tears (Johnson & Johnson Vision, Santa Ana, CA, USA)	PEG 400 HA	Calcium chloride, magnesium chloride, potassium chloride, sodium chloride, boric acid, sodium borate	Preservative: Sodium chlorite (OcuPure^®^)	[29]

Abbreviations: CMC = Carboxymethylcellulose; HA = Hyaluronic acid; HP-guar = Hydroxylpropyl-guar HPMC = Hydroxypropyl methylcellulose; PEG = Polyethylene Glycol; PQ = Polyquaternium-1; PVA = Polyvinyl alcohol.

**Table 2 ijms-23-02434-t002:** Characteristics of selected natural and synthetic viscosity-enhancing agents used in tear substitutes.

Name	Type (Source)	Molecular Weight Range
Carbomer^®^ (polyacrylic acid)	Synthetic polymer	From ~1 kDa to ~3 MDa
Carboxymethyl cellulose	Natural PS (cellulose derivative, from plants)	From ~90 to ~250 kDa
Dextran	Natural PS (glucose derivative)	From ~3 to ~2000 kDa
Hyaluronic acid	Natural exo-PS (from bacterial fermentation)	From ~8 kDa to ~1.8 MDa
Hydroxypropyl methylcellulose	Natural PS (cellulose derivative from plants)	From ~10 to ~120 kDa
Hydroxypropyl guar	Natural PS (guar gum derivative, from plants)	
Polyethylene glycol	Synthetic polymer	From ~1 to ~8 kDa
Polyvinyl alcohol	Synthetic polymer	From ~9 to ~200 kDa
Povidone (polyvinylpyrrolidone)	Synthetic polymer	From ~10 kDa to ~1 MDa
Propylene glycol	Synthetic polymer	From ~1 to ~15 kDa
Tamarind seed PS	Natural PS (from tamarind kernel powder)	From ~400 kDa to ~6 MDa

Abbreviation: PS = Polysaccharide.

**Figure 2 ijms-23-02434-f002:**
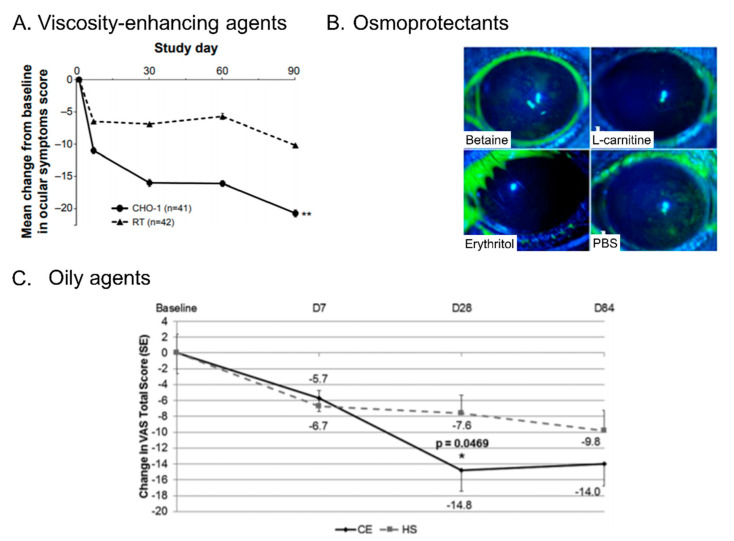
Main outcomes of ingredients contained in marketed artificial tears. (**A**) Mean changes in ocular signs scores of the Ocular Surface Disease Index in patients with DED and treated with ATs containing CMC and HA (CHO-1) or CMC only (RT). ** *p* < 0.007 for CHO-1 versus RT [23]. (**B**) Representative images of CFS of mice after 35 days of topical treatment with betaine, L-carnitine, erythritol or PBS [30]. (**C**) Change from baseline of ocular discomfort score at day 7, 28 and 84. CE = cationic emulsion; HS = Sodium hyaluronate; VAS = visual analogue scale; SE = Standard Error [31].

### 3.2. Electrolytes

Naturally excreted electrolytes constitute the tear film, which maintains the osmotic balance of the ocular surface. For this reason, osmotic agents such as the electrolytes (sodium, potassium, chloride, magnesium and calcium) are largely used in tear substitutes in order to reproduce the electrolyte profile of healthy tear film (Table 1). As shown in the previous section, one of the mechanisms leading to DED is the tear film hyperosmolarity. Several in vivo and clinical studies showed that the use of hypotonic substitutes can reduce signs of DED [32,33,34]. These results may be explained by the dilution of the solute concentration in the tear film causing the hyperosmolarity. It is additionally worth noting that some electrolytes, such as boric acid, can also act as buffering agents to stabilize the pH of formulations or as preservative agents when combined with sorbitol, zinc and propylene glycol (SofZia preservative system) [35,36].

Therefore, electrolytes play an important role in maintaining healthy osmolarity of the tear film, by providing essential ions for the maintenance of the corneal epithelial cells [37] and by counterbalancing the hyperosmolarity of the tear film induced by DED.

### 3.3. Osmoprotectants

The main consequence of hyperosmolarity of the tear film induced by DED is the apoptosis of corneal and conjunctival epithelial cells [7]. In order to prevent this apoptosis, osmoprotectants have been used in some formulations of ATs such as L-carnitine, erythritol, betaine, sorbitol, glycerin and trehalose (Table 3) [38]. L-carnitine and erythritol were found in vitro to protect corneal epithelial cells from hyperosmolar stress [39]. More recently, a combination of betaine, L-carnitine and taurine demonstrated in vitro to protect epithelial cells exposed to hyperosmotic stress [40].

Moreover, some of these osmoprotectants also showed a decreased corneal fluorescein staining (CFS) on an in vivo DED mouse model compared to PBS (Figure 2B) [31]. Trehalose, a natural disaccharide, was also found to protect corneal epithelial cell apoptosis on an in vivo DED mouse model compared to serum and PBS [41]. Clinical studies also showed an improvement of CFS score, without side effects, when patients with moderate to severe DED are treated with trehalose compared with saline [42]. Moreover, trehalose has been shown to protect cells against various threats including inflammation [43] and dysregulated autophagy process [44]. Therefore, trehalose plays an important cytoprotective role for corneal and conjunctival epithelial cells in order to prevent their apoptosis, one of the underlying etiological roles of DED.

It has also been shown that osmoprotectants also reduce matrix metalloproteinase (MMP) synthesis and oxidative stress and can regulate the autophagic process [45,46,47]. Overall, osmoprotectants appear effective in preventing ocular surface cell apoptosis induced by DED.

**Table 3 ijms-23-02434-t003:** Examples of marketed tear substitutes containing osmo-protectants. Depicted composition excludes water and hydrochloric acid/sodium hydroxide for pH adjustment.

Brand Name	Osmo-Protectants	Others	References of Clinical Studies
Thealoz^®^ (Thea Laboratories, Clermont-Ferrand, France)	Trehalose	Viscosity-enhancing agent: HA	[48]
Systane^®^ Ultra (Alcon, Fort Worth, TX, USA)	Sorbitol	Viscosity-enhancing agents: PEG 400, propylene glycol, HP-Guar Electrolytes: potassium chloride, sodium chloride Buffer: aminomethyl propanol Preservative: PQ (POLYQUAD^®^)	[49]
Refresh^®^ Optive^®^ (Allergan, Irvine, CA, USA)	Erythritol, L-carnitine, glycerin	Viscosity-enhancing agent: CMC Electrolytes: Calcium chloride, magnesium chloride, potassium chloride, trisodium citrate dihydrate, sodium borate, sodium chlorate, boric acid Preservative: Sodium chlorite (PURITE^®^)	
Optive Fusion^®^ (Allergan, Irvine, CA, USA)	Erythritol, glycerin	Viscosity-enhancing agent: CMC, HA Electrolytes: Boric acid, sodium borate decahydrate, sodium citrate dihydrate, potassium chloride, calcium chloride dihydrate, magnesium chloride hexahydrate Sodium chlorite (PURITE^®^, preservative)	[23]

Abbreviations: CMC = Carboxymethylcellulose; HA = Hyaluronic acid; HP-guar = Hydroxylpropyl-guar; PEG = Polyethylene Glycol; PQ = Polyquaternium-1.

### 3.4. Oily Agents and Surfactants

As described in Section 2, EDE, characterized by a deficiency of the TFLL, represents the most common type of DED. The presence of lipids and proteins in the lipid layer plays a critical role in the surface tension of the tear film and in the humectation of the ocular surface. Evaporation of tears increases in patients with alterations in the lipid layer [10]. Consequently, oily agents have been used in the formulations of tear substitutes to replenish this layer (Table 4). The oily agents are mainly available in forms of liposomes and oil nanodroplets [50,51].

Liposomes are made of phospholipids forming a spherical vesicle which constitutes one or more concentric lipid bilayers with the same number of aqueous compartments and applied on the ocular surface as a spray. Clinical studies showed that liposomal sprays improved TFLL thickness and tear film stability as well as comfort for the patient [52]. Recently, liposomes have been tested in combination with viscosity-enhancing agents, electrolytes and osmoprotectants in order to more closely resemble the natural tear film [53]. Despite preliminary results showing good tolerance and suitable properties, more studies still need to be performed to assess the performance of this formulation on DED signs.

Another type of tear substitute formulation containing oily agents is oil-in-water emulsions. These emulsions are made of oily droplets stabilized in water using surfactants or emulsifiers. Various types of oils and surfactants have been used to create ophthalmic emulsions [50]. Clinical studies have shown that ATs containing oily agents increased TFLL thickness and patient comfort compared with ATs without oily agents [54,55]. Emulsions can be non-ionic, anionic or cationic according to the components added in the formulation [50]. One interesting feature of cationic emulsions is that the positively-charged oil droplets can interact with the negatively-charged mucin layer of the tear film which helps stabilize the tear film [56,57,58]. The TFLL is naturally composed of both non-polar and polar lipids, and it has been shown that polar lipid abnormalities may be involved in DED [17,59]. Therefore, several formulations have included polar lipid-like surfactants such as cetalkonium chloride (CKC) [57] as well as dimyristoyl-phosphatidylglycerol [60]. Clinical studies highlighted that CEs containing CKC were superior at improving signs in patients with moderate to severe DED, compared with a formulation containing HA [32] (Figure 2C), also exhibiting better improvement of tear break-up time (TBUT) especially in DED patients with MGD [61].

Oily agents and surfactants represent a category of beneficial ingredients which are important for supplementing TFLL, altered due to EDE, the most common category of DED usually induced by MGD. Moreover, oily agents can help provide a smooth optical surface for the cornea and help to maintain a good quality of vision [62].

**Table 4 ijms-23-02434-t004:** Examples of marketed tear substitutes containing lipids and/or surfactants. The composition excludes water and hydrochloric acid/sodium hydroxide for pH adjustment.

Brand Name	Lipids (np: Non-Polar/p: Polar)	Surfactants	Others	References of Clinical Studies
Soothe^®^ XP (Bausch & Lomb, Rochester, NY, USA)	Mineral oils (np)	Octoxynol 40, polysorbate 80	Electrolytes: Boric acid, sodium borate Preservatives: PQ (POLYQUAD^®^), edetate disodium	[63]
Systane^®^ Balance^®^ (Alcon, Fort Worth, TX, USA)	Mineral oil (np) Dimyristoyl phosphatidylglycerol (p)	Polyoxyl 40 stearate	Viscosity-enhancing agents: Propylene glycol, HP-guar Electrolytes: Boric acid Preservatives: PQ (POLYQUAD^®^, edetate disodium	[60]
Cationorm^®^ (Santen, Osaka, Japan)	Mineral oil (np), cetalkonium chloride (p)	Tyloxapol, poloxamer 188,	Osmo-protectants: glycerin Buffers: Tris-HCl, tromethamine	[61]
Cationorm^®^ Pro (Santen, Osaka, Japan)	Medium chain triglycerides (np), cetalkonium chloride (p)	Tyloxapol, poloxamer 188,	Osmo-protectant: glycerin	
Refresh^®^ Digital^®^ (Allergan, Irvine, CA, USA))	Castor oil (p)	Polysorbate 80	Viscosity-enhancing agents: CMC, carbomer Osmo-protectants: Erythritol, L-carnitine, glycerin Preservative: Sodium chlorite (PURITE^®^)	

Abbreviations: CMC = Carboxymethylcellulose; HP-guar = Hydroxylpropyl-guar; PQ = Polyquaternium-1. An-tioxidants.

There is some evidence that DED is also associated with oxidative stress, which induces tissue damage and increases inflammation [64]. Therefore, antioxidants or free radical scavengers have been used in the formulation of tear substitutes including vitamin A, vitamin E, co-enzyme q10 or lipoic acid (Table 5). Erythritol and trehalose which are already used in tear substitute formulations as osmoprotectants, can also protect the cell from oxidative stress [43,65]. A recent study also revealed that taurine can protect corneal epithelial cells from oxidative stress [66]. One clinical study demonstrated that the addition of vitamin A in tear substitutes improved DED signs [67]. However, vitamin A has also been shown to induce Meibomian gland dysfunction in animal models [68]. In addition, vitamin A has been found to be very unstable in liquid formulation, and not well tolerated when used in ointment due to the presence of lanoline. The use of lipoic acid as an antioxidant has also been evaluated for DED patients and may improve tear film stability [69,70]. Moreover, epsilon amino caproic acid, vitamins E, B6 and B12, and panthenol have been largely used in Japanese formulations of tears substitutes [71] (Table 5).

Other clinical studies assessed the topical use of antioxidants, however, they were not tested alone and were used in combination with other components [72,73]. Therefore, more clinical studies are still required to assess the effects of antioxidants alone in artificial tears, in order to better realize their benefits for patients with DED.

**Table 5 ijms-23-02434-t005:** Examples of marketed artificial tears containing antioxidants. The composition excludes water and hydrochloric acid/sodium hydroxide for pH adjustment.

Brand Name	Antioxidants	Others	References of Clinical Studies
VisuXL^®^ (Visufarma, Roma, Italy)	Co-enzyme q10, vitamin E	Viscosity-enhancing agent: HA	[72]
Optrex^®^ Actimist^®^ (Optima Pharmazeutische GmbH, Moosburg an der Isar, Germany)	vitamin A palmitate, vitamin E	Electrolyte: Sodium Chloride Lipids: soy lecithin (polar) Solvents: ethanol, phenoxyethanol	[74]
Neovis Total Multi (Horus Pharma, Saint Laurent du Var, France)	Lipoic acid	Viscosity-enhancing agent: HA, HPMC Electrolyte: Sodium citrate Lipids: Triglycerides (non-polar), phospholipids (polar)	
Lion smile 40 EX a (Lion, Tokyo, Japan)	Vitamin A palminate, Vitamin E Vitamin B6	Others: Potassium L-aspartate, chondroitin sulfate sodium, taurine, neostigmine methylsulfate, chlorpheniramine maleate, tetrahydrozoline hydrochloride, Epsilon aminocaproic acid	
Sante 12 (Santen, Osaka, Japan)	Vitamin B12 Vitamin B6	Others: Potassium L-aspartate, chondroitin sulfate sodium, taurine, neostigmine methylsulfate, chlorpheniramine maleate, tetrahydrozoline hydrochloride, Epsilon aminocaproic acid, panthenol, dipotassium glycyrrhizinate, zinc sulfate hydrate	[73,75]
Rohto Cool 40α (Rohto, Osaka, Japan)	Vitamine E Vitamin B6	Others: Potassium L-aspartate, chondroitin sulfate sodium, neostigmine methylsulfate, chlorpheniramine maleate	

Abbreviations: HA = Hyaluronic Acid; HPMC = Hydroxypropyl methylcellulose Preservatives.

Unlike single dose units of tear substitutes, classic multi-dose units typically require the use of preservatives to prevent microbial growth within the bottle and increase their shelf life. The recognized toxicity of benzalkonium chloride (BAK), the most commonly used preservative in eye drop solutions, led to the development of “soft” preservatives including polyquaternium-1 (PQ, POLYQUAD^®^), sodium chlorite (PURITE^®^ or OcuPURE^®^), edetate disodium (EDTA) (Table 1, Table 3, Table 4 and Table 5) [4]. It has been shown that PQ induced significantly much less in vitro cytotoxicity compared with BAK [76]. However, the absence of preservatives has been shown to improve ex vivo corneal wound healing, which was not observed even with the use of soft preservatives such as PURITE^®^ [77]. Therefore, more clinical studies are needed to evaluate the properties of these soft preservatives.

### 3.5. Agents Promoting Wound Healing and Reducing Inflammation

In addition to its viscosity, HA, especially high molecular weight HA (HMW-HA), has also been shown in vivo to accelerate wound healing of the epithelium after corneal debridement abrasion and alkali burn injuries [15,24,78,79] (Figure 3A). This outcome is most likely due to the presence of ligands on the HMW-HA molecule that can bind to the CD44 receptor present on most cell types, including human corneal epithelial cells (HCEC) [80]. Similarly, CMC has been found to bind HCEC and promote in vivo corneal reepithelialization [81]. Several studies highlighted that HA can also exhibit anti-inflammatory properties via TLR2 and TLR4 regulation in other tissues [82,83,84]. It was shown that HMW-HA can promote a higher TBUT and better reduce ocular surface cell apoptosis when compared with low molecular weight (LMW-HA) [85].

Recently, HA has also been combined with other of agents in order to improve epithelial cell reepithelialization. T-LysYal, a supramolecular system composed of lysine HA, thymine, and sodium chloride was investigated in in vitro studies and showed its potential to restore corneal cells damaged by DED and exhibited anti-inflammatory activity [86,87]. These outcomes showed that T-LysYal acts as an efficient ocular surface modulator and could represent a promising component of ATs. However, in vivo studies are still necessary to confirm these results.

As shown in a previous section, cationic emulsions (CEs) have been found to be beneficial in replenishing the TFLL. A study assessed the effect of CEs on the wound closure after cell culture scraping and results showed that CEs also favored in vitro corneal wound healing [88]. Moreover, in vitro studies revealed that CEs reduced the expression and secretion of pro-inflammatory factors, such as interleukin 6 and 8 from corneal epithelial cells via Protein Kinase C pathway inhibition, suggesting an anti-inflammatory potency of CEs [89] (Figure 3B).

These results suggest that certain ingredients, such as trehalose, HA (especially HMW-HA) and CEs, can have more than one beneficial role in the formulation of ATs. These additional roles can help with the etiological causes of DED, including ocular surface cell apoptosis and inflammation.

**Figure 3 ijms-23-02434-f003:**
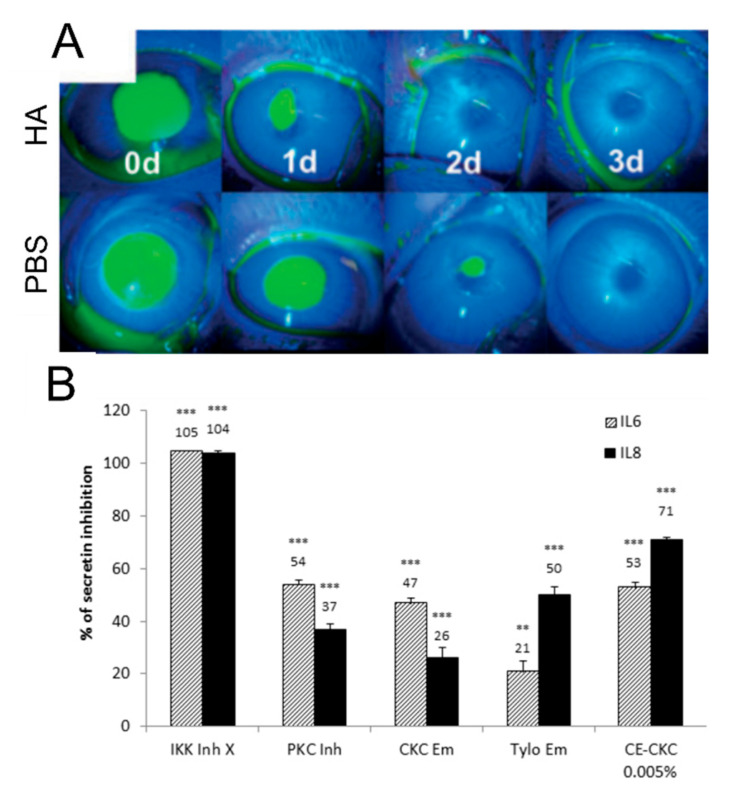
Agents promoting wound healing and reducing inflammation contained in marketed tear substitutes. (**A**) Representative image of rabbit corneal fluorescein staining after epithelial abrasion and treatment with hyaluronic acid (HA) or PBS [79]. (**B**) Effects of emulsion (Em) containing cetalkonium chloride (CKC) or tyloxapol (tylo) and CE on IL-6 and IL-8 secretion inhibition following lipopolysaccharide (LPS) stimulation in cultured HCEC. ** *p* < 0.01; *** *p* < 0.001 [89].

## 4. Considerations of Ingredients for the Management of DED

In this section, we will develop how the described roles (mechanical and biological) can target the different signs involved in the vicious circle of DED (Figure 4).

First, viscosity-enhancing agents have been found to be efficient at replenishing and stabilizing the aqueous layer of the tear film. Moreover, the addition of electrolytes allows better reproduction of the composition of the natural aqueous layer. Tear substitutes containing both of these standard categories of ingredients can thus enter the “wetting agents” category of *Barabino* updated classification [4]. Second, the use of oily agents and surfactants have been found to be beneficial to supplement and stabilize the lipid layer of the tear film, which is deficient for most patients with DED (i.e., EDE) [90]. Third, the use of osmoprotectants has been found to be beneficial to counteract the hyperosmolarity of the tear film, with hypotonic tear substitutes representing another strategy to correct the osmolarity. Tear substitutes containing lipids or osmoprotectants can be considered as “multi-action tear substitutes” since they have more outcomes than simply replenishing the TFAL. Fourth, antioxidants can be used to prevent cell apoptosis caused by oxidative stress, even though more clinical studies need to be performed to prove their efficacy on DED signs. Fifth, CEs and HA have been shown to reduce expression and secretion of pro-inflammatory factors and thus, appear adapted to reduce ocular inflammation induced by DED and to promote wound healing. Therefore, tear substitutes composed of these ingredients can be considered as “ocular surface modulators”, even if in vivo studies should be performed to validate this classification. Consequently, these ingredients may be particularly beneficial for patients with DED signs following surgery. Ingredients composing tear substitutes are thus able to target all the main etiological causes of DED.

The choice and concentration of each ingredient should be carefully selected to provide a safe and efficient product for the patient. On one hand, electrolytes can be beneficial for patients with ADDE in order to reproduce the natural aqueous layer. On the other hand, the use of electrolytes increases the osmolarity of ATs which can potentially exacerbate (or at best, not reduce) the hyperosmolarity of tear film. The use of oily agents has been found to be effective for patients with EDE. However, high concentrations of certain surfactants, which are required to solubilize the oily agents, have shown potential complications including ocular toxicity. Therefore, the type and concentrations of surfactants should be carefully considered to stabilize the TFLL to avoid toxic effects. Finally, only small amounts of preservatives or only “soft preservatives” should be used in tear substitute formulations.

## 5. Conclusions

In addition to the burden on the patients’ quality of life, DED also represents a significant economic burden [1]. Despite the large variety of ATs available on the market, patient satisfaction, driven by resolution of their DED symptoms, remains very low [91]. A better understanding of the roles of ingredients contained in ATs could help provide each patient with the most appropriate formulations, to better target her/his symptoms, in addition to the DED signs [92]. In this review, we have demonstrated that ingredients of tear substitutes not only have significant effects on symptomatic relief, but they can also play a significant therapeutic role in the underlying mechanisms involved in DED. Moreover, some ingredients, such as HA, trehalose and CEs have demonstrated multiple actions and thus, appear particularly adapted for the management of DED. This review also highlighted that marketed ATs are not all equal, and that some formulations are more adapted to specific DED types.

Nowadays, tear substitutes are usually designated as “medical devices” in Europe or “ophthalmic drug products for over-the-counter human use” in the United States. Contrary to medicated eye drops, ATs under these designations may require a lower level of clinical evidence to obtain marketing authorization. For example, the U.S. FDA guidance does not require clinical studies for ATs if they contain specific ingredients described in the OTC monograph [2]. Tear substitutes in the US are usually designated based on their “active ingredients” or “biologically active ingredients” [92]. Conversely, the pharmacological contribution of ingredients is not assessed for ATs with medical device status whereby the claimed mechanism of action is predominantly mechanical (e.g., lubrication, moisturizing…). This review suggests that most of the ingredients have a mechanical and/or biological activity on pathophysiology of DED and that the distinction between inactive versus active ingredients may appear obsolete for some of these excipients in regard to the outcomes described in this review. ATs can also be used in combination with drugs, such as HA with hydrocortisone or CEs with cyclosporine A [93,94]. This shows that ATs ingredients can also be used as functional vehicles for drug delivery on the ocular surface to better improve DED signs and symptoms.

Additional clinical studies (as is now requested by the new European regulation on medical devices) would allow better comparisons of tear substitutes to assess their safety and performance on the different categories of patients with DED (e.g., ADDE, EDE or both) in addition to preclinical studies to better understand pharmacological activity.

## Figures and Tables

**Figure 1 ijms-23-02434-f001:**
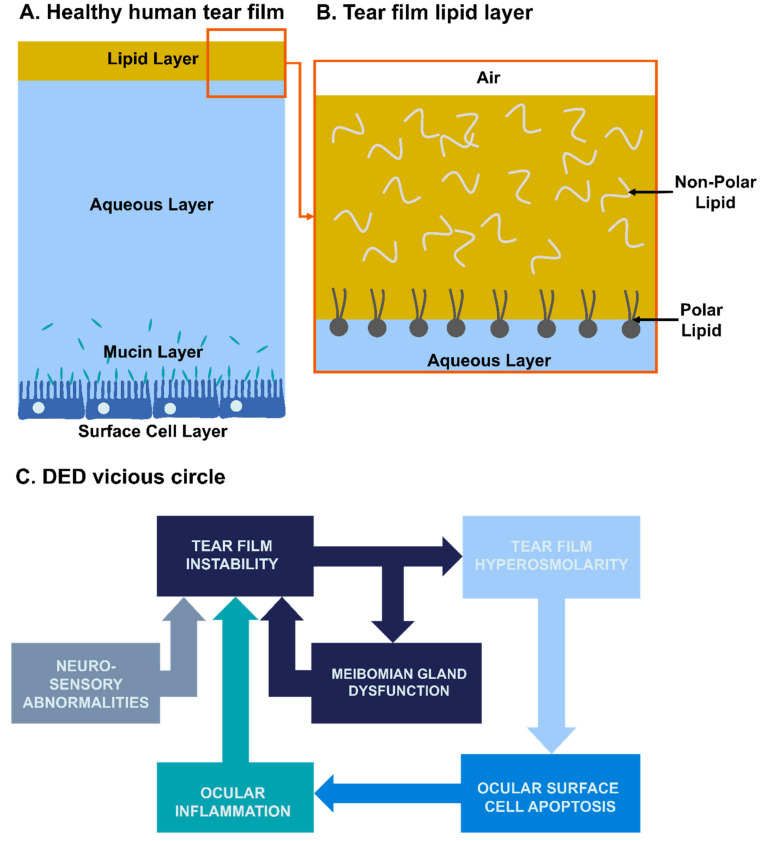
Structure of the tear film and simplified pathogeny of DED. (**A**) Composition of a healthy human tear film, (**B**) zoom on the lipid layer composition and (**C**) Mechanisms involved in the DED vicious circle.

**Figure 4 ijms-23-02434-f004:**
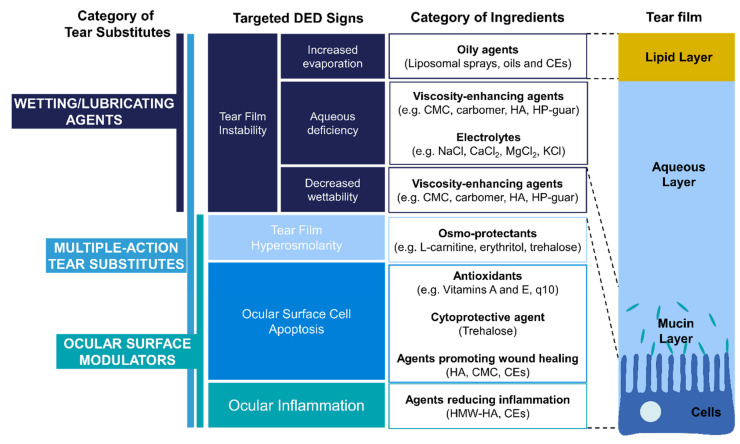
Category of ingredients of artificial tears targeted DED signs. ADDE = Aqueous Deficiency Dry Eye; CEs = Cationic emulsions; CMC = Carboxymethylcellulose; EDE = Evaporative Dry Eye; HA = Hyaluronic acid; HMW = high molecular weight; HP-guar = hydroxypropyl guar.

## Data Availability

Not applicable.

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
