# Peer review of "Artificial Tears: Biological Role of Their Ingredients in the Management of Dry Eye Disease"

_ijms, 2022, doi:10.3390/ijms23052434_

Round 1

Reviewer 1 Report

This is a comprehensive review on the artificial tears with an important impact on the current ophthalmological practice, since DED is an increasing problem. It would be even more useful to add a section presenting the entities causing DED in relation to the component of the tear film that it mainly affects.

Author Response

December 17th, 2021

We would like to thank the reviewers for the time they spent to review our work. Thanks to their valuable comments, we believe that our manuscript has been substantially improved, and we hope that it meets with your approval.

We didn’t not obtain permission for copyright for the Figure 3C so we had to remove it.

The modifications made in the revised manuscript have been highlighted in yellow.

Reviewer #1:

This is a comprehensive review on the artificial tears with an important impact on the current ophthalmological practice, since DED is an increasing problem. It would be even more useful to add a section presenting the entities causing DED in relation to the component of the tear film that it mainly affects.

Response: We thank the reviewer for this valuable comment. The section 4 of the manuscript, and especially the Figure 4, describes the link between each component of the artificial tears with the components of the tear film and the DED symptoms.

Reviewer 2 Report

The authors of this review described the current knowledge base of various components in Artificial Tear products.  They described the biologic effects/causes of the 3 types of DED, as well as going into the current literature on the effects of the various AT components; broken down into the various categories.

Specific comments: 

In section 2, it would be appropriate to spell out TFLL as tear-film lipid layer.  Currently, the authors only say lipid layer.

In section 3, the numbering of sections is wrong.  After section 3.2, all of the sections are listed as 3.3

In the last paragraph of section 3.3 osmoprotectants, the authors claim that osmoprotectants "also reduce MMP synthesis and oxidative stress and can regulate autophagic process."  However, the reference listed, 45, only describes autophagic process regulation, not MMP synthesis or OS.  Additional references required.

In first paragraph of section 3.3 Agents promoting wound healing, this sentence seems conceptually incorrect. "This outcome is most likely due to the presence of CD44 receptor on the HMW-HA molecule"  I would look into this wording, as it seems CD44 receptors would be on the epithelial cells, and HMW-HA molecules would posess the ligand, or binding motif.

In same paragraph, a line reads "SEVERAL studies highlighted that HA can also exhibit anti-inflammatory properties via TLR2 and TLR4 regulation."  Only one reference, not several, is listed (80) and this reference is regarding HA in joints and cartilage.  Either additional references are needed, ideally referencing any studies in eyes with HA and TLR2/4 regualtion, or change the sentence to remove SEVERAL, and make sure to mention the study dealt with HA and joint cartilage.  Seems misleading.

In it's current form, there is a large space before Figure 3, and Figure 3 is oddly shaped; lots of empty space.

Other than the above comments, I think the review is well written and goes into detail describing the current components of various AT products and how each funstions.  The authors do well describing their "ideal" formulations, noting that further study would be required for each of these combined products.

Author Response

December 17th, 2021

We would like to thank the reviewers for the time they spent to review our work. Thanks to their valuable comments, we believe that our manuscript has been substantially improved, and we hope that it meets with your approval.

We didn’t not obtain permission for copyright for the Figure 3C so we had to remove it.

The modifications made in the revised manuscript have been highlighted in yellow.

Reviewer #2:

The authors of this review described the current knowledge base of various components in Artificial Tear products.  They described the biologic effects/causes of the 3 types of DED, as well as going into the current literature on the effects of the various AT components; broken down into the various categories.

Specific comments:

In section 2, it would be appropriate to spell out TFLL as tear-film lipid layer.  Currently, the authors only say lipid layer.

Response: We thank the reviewer for noticing this, we spelled out TFLL as tear-film lipid layer.

In section 3, the numbering of sections is wrong.  After section 3.2, all of the sections are listed as 3.3

Response: We thank the reviewer for this comment, we revised the numbering of sections.

Dear editorial board, please note that the section numbering was correct in the version we submitted.

In the last paragraph of section 3.3 osmoprotectants, the authors claim that osmoprotectants "also reduce MMP synthesis and oxidative stress and can regulate autophagic process."  However, the reference listed, 45, only describes autophagic process regulation, not MMP synthesis or OS.  Additional references required.

Response: We thank the reviewer for this valuable comment, we added the 2 references below to support these claims:

  • Deng R, Su Z, et al. Osmoprotectants suppress the production and activity of matrix metalloproteinases induced by hyperosmolarity in primary human corneal epithelial cells. Mol Vis. 2014 Sep 12;20:1243-52.
  • Luyckx J, Baudouin C. Trehalose: an intriguing disaccharide with potential for medical application in ophthalmology. Clin Ophthalmol. 2011;5:577-81. doi: 10.2147/OPTH.S18827. Epub 2011 May 10.

In first paragraph of section 3.3 Agents promoting wound healing, this sentence seems conceptually incorrect. "This outcome is most likely due to the presence of CD44 receptor on the HMW-HA molecule"  I would look into this wording, as it seems CD44 receptors would be on the epithelial cells, and HMW-HA molecules would posess the ligand, or binding motif.

Response: We thank the reviewer for this great comment, we modified the sentence:

“This outcome is most likely due to the presence of ligands on the HMW-HA molecule that can bind to the CD44 receptor present on most cell types, including human corneal epithelial cells (HCEC)”

In same paragraph, a line reads "SEVERAL studies highlighted that HA can also exhibit anti-inflammatory properties via TLR2 and TLR4 regulation."  Only one reference, not several, is listed (80) and this reference is regarding HA in joints and cartilage.  Either additional references are needed, ideally referencing any studies in eyes with HA and TLR2/4 regualtion, or change the sentence to remove SEVERAL, and make sure to mention the study dealt with HA and joint cartilage.  Seems misleading.

Response: We thank the reviewer for this great comment, we added the 2 references below to support this claim and we added that these studies refer to non-ocular tissues.

Jiang, Dianhua et al. “Regulation of lung injury and repair by Toll-like receptors and hyaluronan.” Nature medicine vol. 11,11 (2005): 1173-9.

Jiang, Dianhua et al. “Hyaluronan as an immune regulator in human diseases.” Physiological reviews vol. 91,1 (2011): 221-64.

In its current form, there is a large space before Figure 3, and Figure 3 is oddly shaped; lots of empty space.

Response: We thank the reviewer for noticing this.

Other than the above comments, I think the review is well written and goes into detail describing the current components of various AT products and how each functions.  The authors do well describing their "ideal" formulations, noting that further study would be required for each of these combined products.
